# Structural analysis of cancer-relevant TCR-CD3 and peptide-MHC complexes by cryoEM

Kei Saotome [1] ✉, Drew Dudgeon [1], Kiersten Colotti[1], Michael J. Moore[1], Jennifer Jones[1], Yi Zhou[1], Ashique Rafique [1], George D. Yancopoulos[1], Andrew J. Murphy [1], John C. Lin[1], William C. Olson[1] & Matthew C. Franklin [1] ✉

The recognition of antigenic peptide-MHC (pMHC) molecules by T-cell receptors (TCR) initiates the T-cell mediated immune response. Structural characterization is key for understanding the specificity of TCR-pMHC interactions and informing the development of therapeutics. Despite the rapid rise of single particle cryoelectron microscopy (cryoEM), x-ray crystallography has remained the preferred method for structure determination of TCR-pMHC complexes. Here, we report cryoEM structures of two distinct full-length α/β TCR-CD3 complexes bound to their pMHC ligand, the cancer-testis antigen HLA-A2/MAGEA4 (230–239). We also determined cryoEM structures of pMHCs containing MAGEA4 (230–239) peptide and the closely related MAGEA8 (232–241) peptide in the absence of TCR, which provided a structural explanation for the MAGEA4 preference displayed by the TCRs. These findings provide insights into the TCR recognition of a clinically relevant cancer antigen and demonstrate the utility of cryoEM for high-resolution structural analysis of TCR-pMHC interactions.

Recognition of pathogenic and cancerous peptide-MHC (pMHC) antigens by T-cells is mediated by T-cell receptors (TCR)[1,2]. TCRs are expressed as heterodimers of α/β or γ/δ chains in complex with three CD3 dimers (CD3εδ, CD3εγ, CD3ζζ) that are responsible for initiating downstream signaling[3]. Sequence diversity in the variable domains, generated by V/D/J recombination similar to immunoglobulins, allow TCRs to discriminate their cognate pMHC molecules from the rest of the MHC-displayed proteome[4]. TCRs that specifically target tumor antigens serve as the basis for soluble and cellular TCR-based cancer immunotherapies that have shown clinical promise[5–9]. Notably, a bis-pecific T-cell redirecting fusion protein that uses an affinity-enhanced TCR specific for an HLA-A2-presented gp100 peptide was recently approved by the FDA for metastatic uveal melanoma[10,11]. In addition, therapeutic cancer vaccines employ MHC-displayed peptides to induce anti-cancer T cell responses[12,13].

Crystallographic studies, spanning over 25 years, have shed light on the structural basis of TCR-pMHC recognition and its relation to the T-cell immune response[14–16]. These studies, using soluble ectodomain proteins, have shown that TCRs use three complementarity-determining regions (CDRs) on each chain to make contacts with the pMHC molecule. Almost all TCR structures have shown a canonical docking mode in which the CDR1 and CDR2 loops interact primarily with the MHC molecule and CDR3 loops contact the MHC-embedded peptide, governing antigen recognition[17]. Elucidating the structural basis of antigen specificity is of particular interest for TCRs with therapeutic potential because off-target reactivity to peptides presented on healthy cells can have dangerous consequences[18]. For example, crossreactivity of an anti-MAGEA3 TCR T-cell therapy to a peptide derived from Titin, expressed in cardiac tissue, resulted in two deaths during a clinical trial[19]. Subsequent structural studies showed that the Titin peptide closely mimicked the conformation of the MAGEA3 peptide within the MHC groove despite having sequence differences at 4 of 9 residues, allowing TCR crossreactivity[20]. In principle,

[1]Regeneron Pharmaceuticals, Inc., Tarrytown, NY 10591, USA. ✉e-mail: kei.saotome@regeneron.com; matthew.franklin@regeneron.com

structural information can help improve the safety and efficacy of TCR-based therapeutics by facilitating predictions of off-target peptides[21] and structure-guided enhancement of TCR-pMHC interaction[22–24].

Structures of TCR-pMHC complexes reported thus far have almost exclusively been solved by x-ray crystallography using soluble ectodomain reagents. Successful crystallization of these complexes remains a difficult task, notwithstanding advances in engineered protein constructs[25], expression strategies[26], and crystallization screens[27]. Notably, a landmark cryoelectron microscopy (cryoEM) structure of a full-length TCR-CD3 complex was recently described[28], as well as the first structure of a full-length affinity-enhanced TCR-CD3 bound to pMHC[29]. However, the application of cryoEM towards TCR-pMHC complex structure determination remains in its infancy. Furthermore, whether the naturally low affinity of the TCR-pMHC interaction precludes resolution of cryoEM structure remains unknown.

Here, we use cryoEM to investigate TCR-pMHC recognition, focusing on two μM affinity α/β TCRs derived from humanized mice[30] that target a peptide epitope containing residues 230–239 from the cancer-testis antigen MAGEA4. This peptide is presented by the MHC molecule HLA-A2 in numerous solid tumors, but not healthy adult tissues, making it an attractive target for TCR-based therapies and cancer vaccines[31–33]. We determined cryoEM structures of two full-length TCR-CD3 complexes bound to HLA-A2/MAGEA4 (230–239), as well as structures of HLA-A2/MAGEA4 (230–239) and the closely related HLA-A2/MAGEA8 (232–241) pMHCs in the absence of TCR. Our results elucidate how a key cancer antigen is recognized by two novel TCRs and suggest a structural mechanism for preferential binding of the two TCRs to MAGEA4 over MAGEA8. Furthermore, our study demonstrates that cryoEM is suitable for determining high-resolution structures of MHC antigens and their complexes with TCR.

## Results

### CryoEM structure of full-length PN45545 TCR-CD3

Two α/β TCRs, PN45545 and PN45428, were isolated from humanized VelociT mice[30] immunized with the MAGEA4 (230–239) peptide. We first focused our studies on the PN45545 TCR. Adapting previous approaches[28,34], we expressed a full-length PN45545 TCR-CD3εδγζ complex in HEK293 cells and purified it to homogeneity in detergent without chemical crosslinking (Supplementary Fig. 1). We determined a cryoEM structure of the PN45545 TCR-CD3 complex to 3.0 Å resolution (Fig. 1a, b Supplementary Fig. 2a–c, Table 1). Side chain densities were well resolved for most residues in the extracellular (ECD) and transmembrane (TM) domains of each subunit in the complex (Supplementary Fig. 2d). N-linked glycan densities were identified on TCRα (N58, N150, N184, N195), TCRβ (N84, N107, N184), CD3δ (N38, N74), and CD3γ (N52, N92) (Fig. 1b). We also noted the presence of a lipid density situated between the TM helices of TCRβ, CD3γ, and CD3ζ subunits that we tentatively assigned as cholesteryl hemisuccinate (CHS) due to its matching shape features and its presence in purification buffer (Fig. 1a, c, Supplementary Fig. 2e). A cholesterol-like density was observed at this location in two recent cryoEM studies, one of which proposed a functional role for this lipid[29,35]. Interestingly, the cryoEM map suggests the possibility of S-palmitoylation at CD3δ residue C124 (Supplementary Fig. 2f), consistent with a palmitoylation profiling study that revealed CD3δ as a high-confidence target[36]. The cytoplasmic tails of the CD3 subunits were not resolved in the cryoEM map, presumably due to flexibility. The structure and arrangement of the TCR constant regions and CD3 subunits is nearly identical to previously published TCR-CD3 complex structures[28,29] (Fig. 1c–e). However, the elbow angle between TCR variable and constant regions is slightly different, likely reflecting their distinct Vα/Vβ sequences (Fig. 1e). Taken together with previous cryoEM studies that focused on different α/β TCR pairs[28,29,35], our PN45545 TCR-CD3 complex structure supports the notion that the overall structure and assembly of TCR-

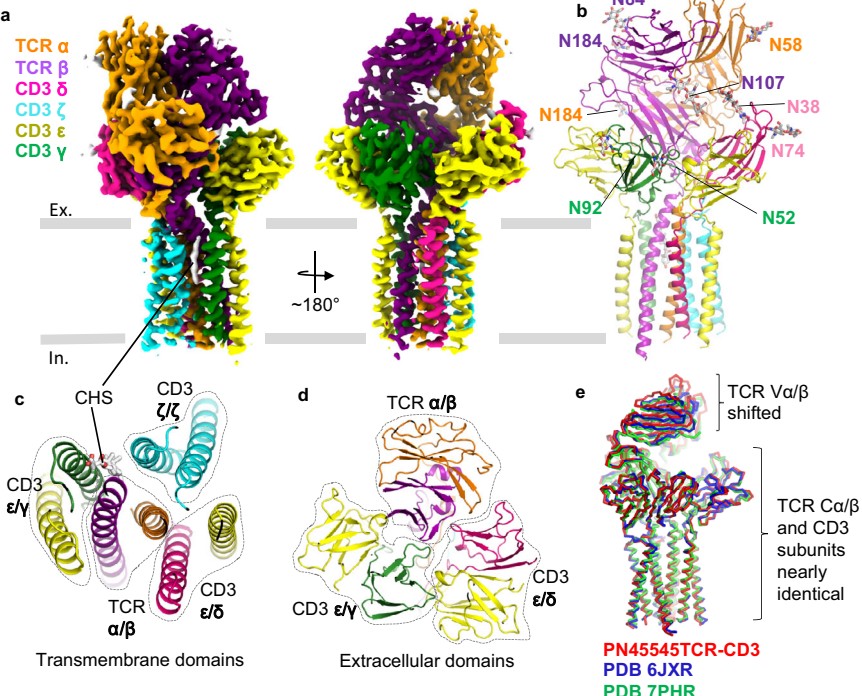

**Fig. 1 | CryoEM structure of PN45545-TCR CD3 complex. a** Two views of 3.0 Å resolution cryoEM map of PN45545 TCR-CD3, with subunits in different colors. **b** Structure of PN45545 TCR-CD3, with N-linked glycans shown in stick representation. **c** top-down view of transmembrane domains. Putative CHS (cholesteryl hemisuccinate) molecule is shown as sticks. **d** top-down view of extracellular domains. In **c** and **d**, the TCR αβ and CD3 εδ/εγ/ζζ dimers are encircled by dotted lines. **e** structural alignment of PN45545 TCR-CD3 (red ribbon) and reported TCR-CD3 structures (blue and green ribbons). The bound pMHC molecule was removed from PDB 7PHR for clarity.

**Table 1 | CryoEM data, structure refinement, and validation**

| | PN45545 TCR-CD3 | PN45545 TCR-CD3 in complex with HLA-A2 MAGEA4 (230-239) | PN45428 TCR-CD3 in complex with HLA-A2 MAGEA4 (230-239) | HLA-A2 MAGEA4 (230-239) | HLA-A2 MAGEA8 (232-241) |
|---|---|---|---|---|---|
| **Data collection and processing** | | | | | |
| Magnification | 105,000 | 105,000 | 105,000 | 105,000 | 105,000 |
| Voltage (kV) | 300 | 300 | 300 | 300 | 300 |
| Electron exposure ($e^-/Å^2$) | ~40 | ~40 | ~40 | ~40 | ~40 |
| Defocus range (μm) | −1.4 to −2.4 | −1.4 to −2.4 | −1.4 to −2.4 | −1.4 to −2.4 | −1.4 to −2.4 |
| Pixel size (Å) | 0.85 | 0.85 | 0.85 | 0.85 | 0.85 |
| Number of movies | 4,889 | 6,907 | 5,921 | 4,675 | 4,995 |
| Initial number of particles | 1.9 M | 1.8 M | 3.1 M | 3.4 M | 5.1 M |
| Particles selected after 2D classification | 762 K | 1.1 M | 2.9 M | 751 K | 1.4 M |
| Final selected particles | 137,831 | 228,624 | 107,308 | 76,433 | 121,731 |
| Symmetry imposed | C1 | C1 | C1 | C1 | C1 |
| Map resolution (Å) | 3.04 | 2.65 | 3.25 | 3.40 | 3.12 |
| FSC threshold | 0.143 | 0.143 | 0.143 | 0.143 | 0.143 |
| **Refinement** | | | | | |
| Initial Model used | 6JXR | | | 1I4F | |
| **Model composition** | | | | | |
| Non-hydrogen atoms | 8,822 | 11,977 | 11,845 | 3,142 | 3,142 |
| Protein residues | 1,080 | 1,464 | 1,459 | 384 | 384 |
| Ligands | 19 | 19 | 14 | 0 | 0 |
| **R.m.s. deviations** | | | | | |
| Bond lengths (Å) | 0.003 | 0.003 | 0.004 | 0.004 | 0.006 |
| Bond angles (°) | 0.586 | 0.654 | 0.653 | 0.519 | 0.648 |
| **Validation** | | | | | |
| MolProbity score | 1.77 | 1.56 | 1.75 | 1.59 | 1.58 |
| Rotamer outliers (%) | 0.10 | 0.08 | 0.00 | 0.30 | 0.00 |
| Clash score | 6.76 | 6.28 | 6.77 | 5.74 | 5.09 |
| Ramachandran plot | | | | | |
| Favored (%) | 94.08 | 96.60 | 94.57 | 96.01 | 95.48 |
| Allowed (%) | 5.92 | 3.40 | 5.36 | 3.99 | 4.52 |
| Disallowed (%) | 0.00 | 0.00 | 0.07 | 0.00 | 0.00 |
| **Deposition ID** | | | | | |
| PDB | 8ES7 | 8ES8 | 8ES9 | 8ESA | 8ESB |
| EMDB | 28570 | 28571 | 28572 | 28573 | 28574 |

CD3 is unaffected by differences in TCR variable region sequence. Furthermore, the PN45545 TCR-CD3 structure confirms that antigen-specific TCRs discovered in humanized mice[30] have expected structural features.

**CryoEM structures of full-length PN45545 TCR-CD3 and PN45428 TCR-CD3 complexes with HLA-A2/MAGEA4 (230−239)**

To gain insights into the structural basis of MAGEA4 (230−239) peptide recognition by two distinct TCRs, we determined cryoEM structures of full-length PN45545 and PN45428 TCR-CD3 complexes ligated to HLA-A2/$\beta_2$M/MAGEA4(230−239) (henceforth referred to as MAGEA4 pMHC) to resolutions of 2.7 and 3.3 Å, respectively (Fig. 2, Supplementary Fig. 3,4). These structures were determined in the presence of the Fab fragment of the commercially available anti-$\beta_2$M monoclonal antibody 2M2, which was used as a fiducial marker to improve the cryoEM signal of MAGEA4 pMHC (Fig. 2a, d). The cryoEM maps displayed clear side chain densities for the MAGEA4 peptide and nearby regions, enabling unambiguous model building and assessment of amino-acid level interactions at the TCR-pMHC interface (Fig. 2b, e Supplementary Fig. 3d, e, 4d, e). The only notable differences when comparing the unligated and ligated PN45545 TCR-CD3 structures

were present at the CDRs. However, we suggest cautious interpretation of these structural differences because the CDRs are not well defined in the PN45545 TCR-CD3 map calculated in the absence of pMHC.

The overall TCR-pMHC binding orientations of PN45545 and PN45428 follow the canonical docking polarity important for productive coreceptor binding[2,37], with the Vα regions positioned toward the HLA α2 helix and the Vβ regions positioned toward HLA α1 (Fig. 2c, f). However, the two TCRs engaged pMHC with distinct binding modes, characterized by docking angles[38] of 45° and 94° for PN45545 and PN45428, respectively (Fig. 2c, f, g). To assess CD8 coreceptor binding geometry, we used a published crystal structure of murine CD8αβ bound to MHC[39] to model binding of CD8αβ in the context of both full-length MAGEA4 TCR-CD3 complexes (Supplementary Fig. 5). Despite their distinct docking angles, both PN45545 and PN45428 bind MAGEA4 pMHC such that the CD8αβ Ig domain C-termini would be oriented toward the T cell membrane, favoring a *cis* configuration of TCR/CD3/CD8 and *trans* binding of pMHC/CD8 as proposed previously[29,39]. This observation illustrates how the geometric constraints of coreceptor binding can accommodate a range of TCR/pMHC binding angles.

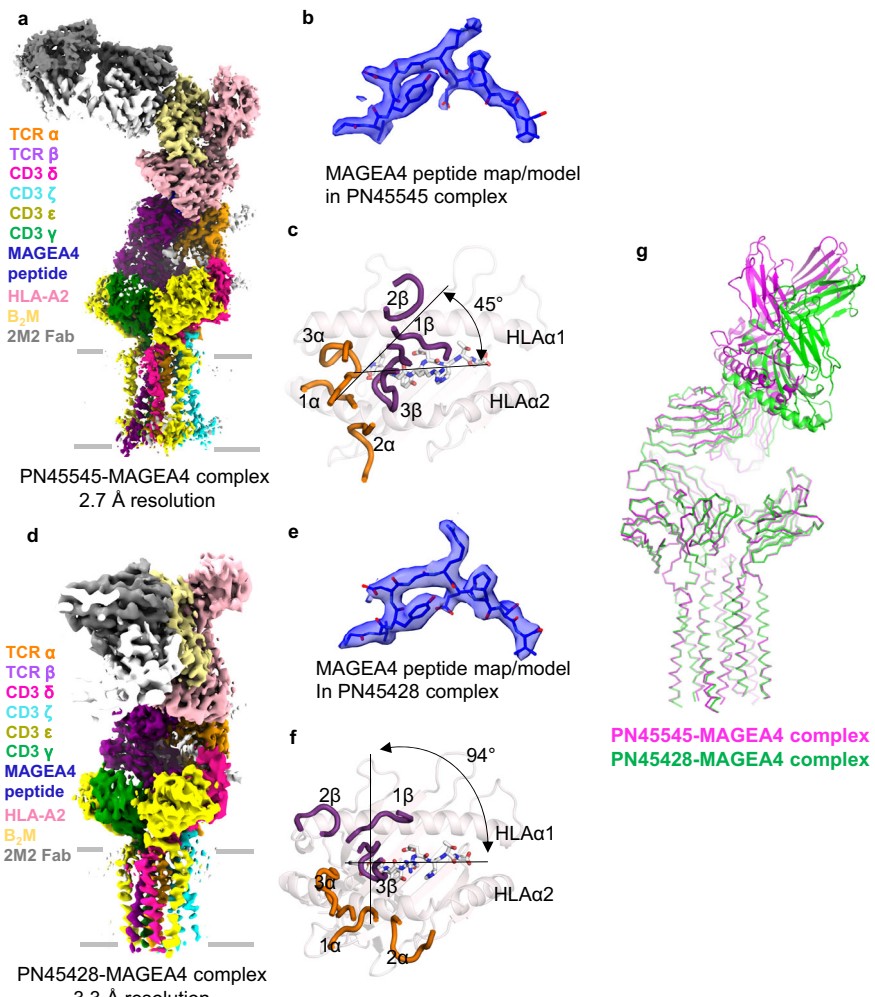

**Fig. 2 | CryoEM structures of TCR-CD3 complexes with MAGEA4 pMHC.**
**a** cryoEM map of PN45545 TCR-CD3 complex with MAGEA4 pMHC. **b** fit of MAGEA4 peptide into the map of the PN45545 complex. **c**, top-down view of MAGEA4 pMHC with α (orange) and β (purple) CDRs of PN45545 shown as loops. TCR interdomain vector and MHC groove vector are depicted as thin lines. The TCR docking angle reported by TCR3d database is shown. **d** cryoEM map of PN45428 TCR-CD3 complex with MAGEA4 pMHC. **e** fit of MAGEA4 peptide into the map (semi-transparent surface) of the PN45428 complex. **f** top-down view of MAGEA4 pMHC with α (orange) and β (purple) CDRs of PN45428 shown as loops. TCR interdomain vector and MHC groove vector are depicted as thin lines. The TCR docking angle reported by TCR3d database is shown. **g** TCR-based structural alignment of PN45545 (magenta) and PN45428 (green) complexes with MAGEA4 pMHC.

Although their docking angles and CDR sequences are distinct (Fig. 3a), both TCRs have a binding footprint that is shifted toward the N-terminus of the MAGEA4 peptide (Fig. 3b–e). In both cases, TCR contacts (within 4 Å interatomic distance) are limited to peptide residues D4, G5, and R6, which form a solvent-exposed bulge[40]. The C-terminal portion of the peptide (E7-V10) is not contacted, suggesting it does not play a direct role in TCR recognition. Notably, a previously reported α/β TCR identified from a healthy donor also displayed an N-terminally shifted binding mode on MAGEA4 pMHC[41], indicating that the N-terminal portion of the peptide and the nearby HLA region may be immunodominant in both humans and humanized mice models.

Peptide contacts for both PN45545 and PN45428 are mediated mainly by CDR3β. In the case of PN45545, CDR3β residues F95 and Y99 make apparent cation-π and hydrogen bond interactions with peptide residues R6 and D4, respectively (Fig. 3d). These and other residues in CDR3β also make van der Waals interactions with peptide G5. CDRs 1α (S31) and 3α (G97) contribute additional contacts to peptide D4 (Fig. 3d). For PN45428, E103 of CDR3β forms a salt bridge with peptide R6, while additional residues in CDR3β contact backbone atoms of peptide G5 and R6 (Fig. 3e). PN45428 α chain residues R31 and N97 make salt-bridge and polar interactions, respectively, with peptide D4 (Fig. 3e). The peptide conformation in the two structures remains

nearly identical, excepting distinct R6 side chain rotamers that are stabilized by the unique TCR-specific interactions (Fig. 3f). The mobility of the R6 side chain, noted previously in a crystal structure of MAGEA4 pMHC without TCR[40], therefore appears to be critical for recognition of this MAGEA4 peptide antigen by different TCRs.

## Structural basis of TCR discrimination between highly similar MAGEA4 and MAGEA8 peptides

HLA-A2/MAGE-A4(230–239) reactive TCRs have previously shown crossreactivity to a similar HLA-A2-restricted peptide derived from MAGEA8 residues 232–241[31]. The MAGEA4 (230–239) and MAGEA8 (232–241) peptides only differ by two conservative substitutions: a valine to leucine replacement at position 2 (V2L), which is buried, and a threonine to serine replacement at the moderately surface-exposed residue at position 9 (T9S). To assess the MAGEA4/MAGEA8 crossreactivities for PN45545 and PN45428, the TCRs were first expressed in primary human T cells and analyzed by flow cytometry with pMHC tetramer reagents. Interestingly, PN45428 showed binding to HLA-A2/MAGEA8 (232–241), though to a lesser degree relative to HLA-A2/MAGEA4 (230–239) (Fig. 4a). However, for PN45545, no binding signal for HLA-A2/MAGEA8 was detected. To confirm the reduced affinity for the MAGEA8 antigen, a steady state

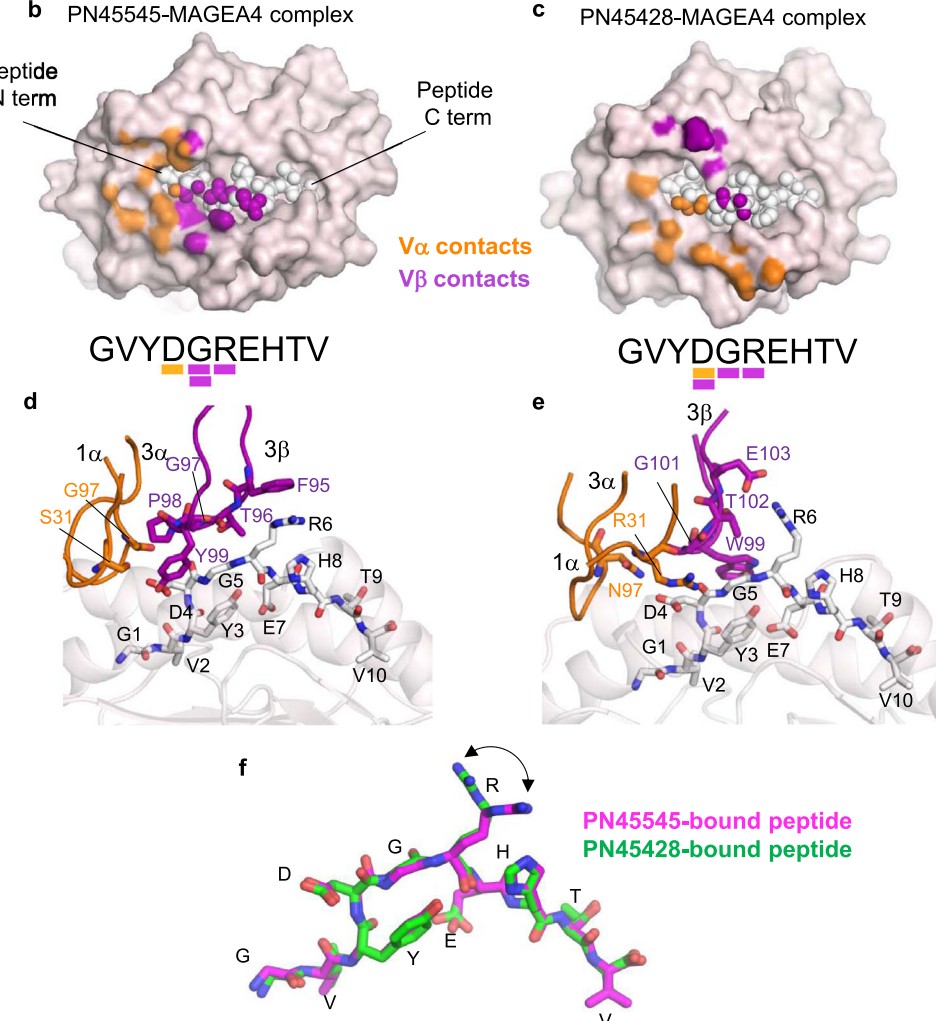

**Fig. 3 | TCR-MAGEA4 pMHC interactions. a** CDR sequences of PN45545 and PN45428 TCRs. **b, c** Top-down view of MAGEA4 pMHC complexes with TCR (PN45545 in **b**, PN45428 in **c**), with HLA shown in surface representation and peptide shown as spheres. Atoms within 4.0 Å of TCR Vα (orange) or Vβ (purple) are colored. MAGEA4 peptide sequence is displayed underneath, with residues contacted by TCR underlined according to the same color scheme as above. **d, e** Expanded views of PN45545 (**d**) and PN45428 (**e**) TCR interactions with MAGEA4 pMHC. CDRs making contacts with peptide are shown as loops and labeled. Amino acids making contacts with peptide are shown as stick and labeled. **f** alignment of MAGEA4 peptides extracted from the PN45545 complex (magenta) and PN45428 complex (green) shows that the central arginine residue (R6) adopts different rotamers.

surface plasmon resonance (SPR) binding assay was used to determine equilibrium binding affinities for the detergent-solubilized TCR-CD3s and MAGEA4/MAGEA8 pMHCs (Fig. 4b, c). PN45428 demonstrated approximately 10-fold tighter affinity for MAGEA4 as compared to MAGEA8 and PN45445 demonstrated approximately 70-fold tighter affinity for MAGEA4, consistent with the greater specificity observed for the PN45545 TCR by flow cytometry (Fig. 4a).

To delineate which of the two substitutions (V2L and T9S) is responsible for preferential binding of the TCRs to MAGEA4 over MAGEA8, we tested binding of pMHC tetramer reagents harboring the singly substituted peptides V2L or T9S to Jurkat cells expressing PN45545 or PN45428 TCRs by flow cytometry (Supplementary Fig. 6). We observed that the V2L mutant showed negligible binding to PN45545 and reduced binding to PN45428 relative to wild type MAGEA4 peptide. The T9S mutant peptide showed binding to both

PN45545 and PN45428 at levels similar to MAGEA4. Therefore, the reduced affinity of the TCRs towards HLA-A2/MAGEA8 can be attributed to the V2L anchor residue substitution.

The preference of both TCRs for MAGEA4 (230–239) over MAGEA8 (232–241) can't be explained directly by the peptide-binding modes of PN45545 or PN45428 (Fig. 3b–e); neither of the two residues that differ between the peptides is contacted by the TCRs. To further investigate the structural differences between these two peptide epitopes, we conducted cryoEM analysis of MAGEA4 and MAGEA8 pMHCs in the absence of TCR. Single-chain disulfide-stabilized pMHC reagents[42] were used for these experiments, and a Fab fragment of anti-β2M antibody 2M2 was added to the samples as a fiducial to facilitate cryoEM data processing. 3.4 and 3.1 Å resolution reconstructions were obtained for MAGEA4 and MAGEA8 pMHCs, respectively (Fig. 5a, b, Supplementary Fig. 7, 8). The maps were sufficiently

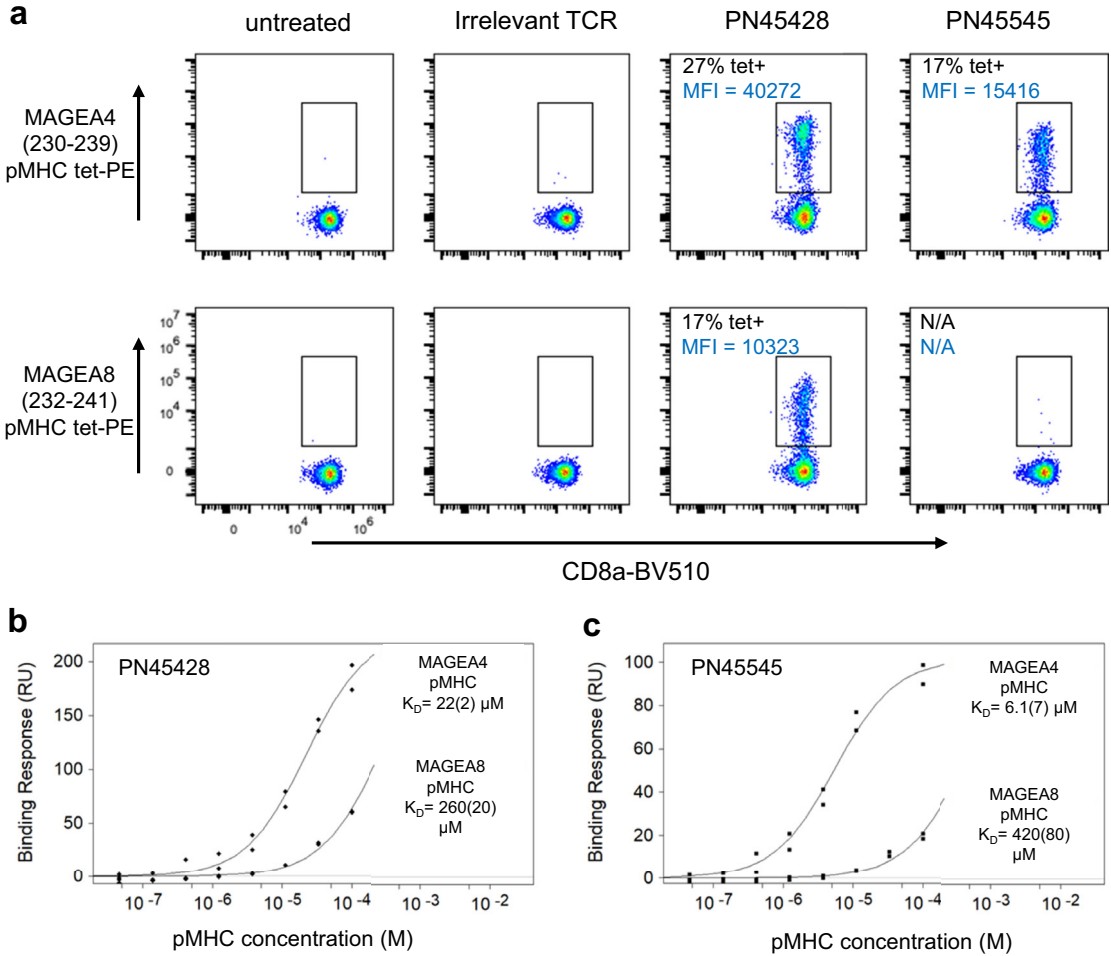

**Fig. 4 | Preferential binding of TCRs to MAGEA4 pMHC over MAGEA8 pMHC. a** Flow cytometry analysis of pMHC tetramers binding to primary human T cells expressing PN45428 or PN45545 TCRs, gated on live, single CD8+ cells. At least 30,000 live, single cells were acquired for each sample. SPR binding responses of full-length TCR-CD3 reagents against MAGEA4 and MAGEA8 pMHC demonstrate

that both PN45428 (**b**) and PN45545 (**c**) preferentially bind to MAGEA4. Steady state KD values are listed. Numbers in parentheses represent SEM (calculated from duplicate experiments on the same sample). Source data for **b** and **c** are provided as a Source Data file.

resolved to define the conformations of HLA-embedded peptides. The side chain of the central R6 residue was not well-resolved (Supplementary Fig. 7d, 8d), consistent with its flexibility in the absence of TCR. As expected from their conserved sequences, the structures of MAGEA4 and MAGEA8 peptides were highly similar (Fig. 5c). The most notable difference occurs at peptide residue D4. The D4 side chain projects down toward the HLA α2 helix in the MAGEA4 peptide, while it protrudes up, away from HLA groove in the MAGEA8 peptide, appearing to form a salt-bridge interaction with the R65 side chain at HLA-α1 (Fig. 5 d–f). The distinct D4 conformations can be attributed to the V/L substitution at position 2; the bulkier leucine side chain in MAGEA8 results in a slight remodeling of the peptide backbone that in turn favors the formation of the peptide D4/HLA R65 salt bridge. The down conformation of peptide D4 is also observed in previously reported crystal structures of MAGEA4 pMHC in the absence[40] and presence of TCR[41], suggesting that it is a relatively stable feature of the MAGEA4 peptide that is perturbed upon introduction of leucine at position 2.

The structures of MAGEA4 and MAGEA8 pMHCs suggest that different conformations of peptide residue D4 are key for discrimination. Indeed, both TCRs make multiple contacts with the peptide D4 side chain, which adopts the down conformation (Figs. 3d, e, 5g, h). Moreover, both TCRs interact directly with HLA-A2 α1 helix residue R65 through residues in CDR2β. Notably, R65 frequently

participates in TCR interactions as part of an HLA 'restriction triad'[43]. For PN45545, Y50 from CDR2β appears to make a cation-π interaction with R65 (Fig. 5g), while in PN45428, E52 from CDR2β forms an electrostatic interaction with R65 (Fig. 5h). These interactions may be compromised in the context of HLA-A2/MAGEA8, where the positive charge of R65 is neutralized by its interaction with peptide D4. Taken together, our structural data indicate that preferential binding by these TCRs for MAGEA4 is likely due to interactions that require the peptide D4 to be in a down conformation where it does not interact with HLA residue 65. Adoption of a MAGEA4-like peptide conformation by MAGEA8 would require disruption of the peptide D4-R65 salt bridge (Fig. 5f), which may be energetically unfavorable.

## Discussion

Here, we used cryoEM to investigate the structural basis for the recognition of the cancer antigen HLA-A2/MAGEA4 by two TCRs (PN45545 and PN45428) isolated from mice with humanized T-cell immunity[30]. Our structures of two full-length, glycosylated TCR-CD3 complexes bound to antigen showed overall canonical antigen docking modes, though they both displayed a distinct shift toward the N-terminal side of the peptide. A remarkable feature of these TCRs is their apparent preference for the MAGEA4 peptide over a highly homologous peptide from MAGEA8. Our structural analysis of MAGEA4 and MAGEA8 pMHCs showed how a conservative valine to

**Fig. 5 | CryoEM structures of MAGEA4 and MAGEA8 pMHCs in complex with 2M2 Fab show distinct MHC-displayed peptide conformations. a, b** cryoEM reconstructions of MAGEA4 and MAGEA8 pMHC complexes with 2M2 Fab, with polypeptide chains shown in different colors. Sequence differences between the two peptides are highlighted red in **b. c, d** two different views of the structural alignment of MAGEA4 (yellow) and MAGEA8 (cyan) peptides embedded in the HLA groove. Expanded view of region around peptide residue D4 in MAGEA4 pMHC (**e**) and MAGEA8 pMHC (**f**). Superimposed cryoEM maps are shown as blue mesh. Expanded top-down views of MAGEA4 pMHC molecule from the PN45545 TCR-CD3 complex (**g**) or the PN45428 TCR-CD3 complex (**h**). TCR residues that directly contact peptide residue D4 or HLA residue R65 are shown as orange (α) or purple (β) sticks.

leucine substitution at anchor residue position 2 can allosterically impact the conformation of solvent-exposed peptide residues contacted by TCR, thus affecting recognition. This finding is consequential for the development of cancer vaccines that use mutated anchor residues (often introducing leucine at position 2) to improve stability of the peptide in the HLA groove[44,45], and corroborates previous studies showing that anchor residue modification can impact TCR recognition[46–49]. Our study therefore highlights the importance of

obtaining structural data to understand the effects of subtle sequence variations on peptide presentation by MHC.

Our complex structures show how TCRs engage pMHC in the context of full-length signaling complexes containing CD3 subunits, allowing for an assessment of the relative positions of the antigen and T-cell membrane that is not possible when using soluble TCRs for structure determination. We note that we did not find any significant structural changes in the TCR constant domains or CD3 subunits

induced by antigen ligation, in agreement with a recently reported structure of an affinity-enhanced TCR-CD3 complex bound to pMHC antigen[29]. The apparent lack of ligand-induced conformational change is consistent with a report showing that multivalent engagement of multiple TCR-CD3 complexes by dimeric or tetrameric pMHC is required to detect a conformational change at CD3ε[50]. TCR triggering by pMHC has also been suggested to require the application of external mechanical force[51]. Finally, pMHC binding may allosterically induce changes in TCR-CD3 dynamics[52], the effects of which would be difficult to observe from static cryoEM structures. The structural mechanisms underlying activation of TCR-CD3 signaling by pMHC thus requires further investigation.

The 'resolution revolution' has resulted in cryoEM becoming the preferred method for structure determination of many classes of macromolecules, in particular membrane proteins and large, flexible complexes that are difficult to crystallize[53,54]. CryoEM has also become an important technique for obtaining structural information for antibody-antigen complexes, as exemplified by numerous recently reported structures of antibody Fab fragments bound to SARS-COV2 spike protein[55,56]. However, TCRs, pMHC antigens, and their complexes have remained in the realm of x-ray crystallography, perhaps due the existence of well-established protocols for producing engineered ectodomain constructs in the milligram quantities required for crystallization[25,26,57,58]. The relatively small molecular size of the soluble components of the complex and their typically low affinities may also present obstacles for cryoEM analysis.

We demonstrate here that cryoEM can yield high-resolution insights into TCR-pMHC recognition, obviating the bottlenecks involved in producing diffraction-quality crystals. Importantly, the TCRs in this study have affinities within the typical μM range (Fig. 4b, c), demonstrating that affinity-matured TCRs, as employed in a recent cryoEM study[29], are not required to stabilize the complex for cryoEM. We also found that the Fab fragment of a commercially available anti-human β$_2$M antibody is an effective fiducial for making class I MHC molecules sufficiently large for cryoEM. This strategy should also facilitate cryoEM analysis of antigen presentation by CD1 and MR1 molecules, which are also noncovalently associated with β$_2$M[59,60]. The additional mass provided by the 2M2 fab will also likely be useful for cryoEM structure determination of TCR-MHC complexes using engineered soluble ectodomain TCR constructs typically used for crystallography. Overall, this study builds on recent reports[28,29,35] to serve as proof of principle for the application of cryoEM in structural studies of TCR-pMHC recognition, which we anticipate will accelerate progress in mechanistic studies and aid in the development of cancer immunotherapies.

## Methods

### TCR-CD3 constructs
TCR-CD3 construct designs were adapted and modified from previous approaches[28,34]. TCR and CD3 DNA constructs were each synthesized in codon-optimized form as single ORFs and cloned into pEZT BacMam[61] and pCAG vectors by GenScript. The full-length PN45545 and PN45428 TCR constructs were comprised of the β chain followed by the α chain with an intervening linker containing a furin cleavage sequence and P2A cleavage site (full amino acid sequence of linker is SRGRAKRGSGATNFSLLKQAGDVEENPGP). The following N-terminal signal sequences were used: MGFRLLCCVAFCLLGAGPV (α chain), MSLSSLLKVVTASLWLGPGI (β chain). The CD3 construct was designed as follows: CD3ε-T2A-CD3γ-P2A-CD3δ-E2A-CD3ζ−3C cleavage site-GFP-strep tag[28,34]. Gly-Ser-Gly linkers were placed N-terminal to each 2A cleavage site and the 3C cleavage site.

### TCR-CD3 expression
Protein used for the structure of PN45545 TCR-CD3 in the absence of antigen was expressed by transient transfection using the constructs cloned into pCAG vectors. 0.4 mg each of TCR and CD3 DNA were mixed with 3 mg of PEI MAX (Polysciences) and added to 0.8 L of HEK293F cells (Thermo Fisher R79007) grown in suspension in Free-Style 293 media (Thermo Fisher). Transfected cells were incubated at 37 °C and 8% $CO_2$ for 24 hours then 1 mM Na butyrate was added and incubated at 37 °C and 8% $CO_2$ for an additional 24 hours. Cells were harvested by centrifugation, washed with PBS, and stored at −80 °C.

Protein used for the structures of PN45545 and PN45428 TCR-CD3 complexes with MAGEA4 pMHC were expressed using BacMam-mediated viral transduction of HEK293F cells. BacMam viruses for TCR and CD3 constructs were produced in Sf9 cells (Thermo Fisher 11496015) maintained in Sf900 II media (Thermo Fisher). P2 viral stocks were concentrated by centrifugation at ~54,000 x $g$ in a Type 70 Ti rotor followed by resuspension of the viral pellet in Freestyle 293 media. HEK293F cells were transduced with a 25% v/v ratio for each virus and incubated at 37 °C and 8% $CO_2$ for 12 hours, at which point 10 mM Na butyrate was added and the temperature was shifted to 30 °C for an additional 36 to 48 hours. Cells were harvested by centrifugation, washed with PBS, and stored at −80 °C.

### TCR-CD3 purification
Cells were thawed and resuspended in buffer containing 50 mM Tris pH 8.0, 150 mM NaCl, 1% Glyco-diosgenin (GDN), 0.15% cholesteryl hemisuccinate (CHS), and EDTA-free cOmplete protease inhibitors (Roche). The mixture was stirred at 4 °C for ~1.5 hours and then clarified by centrifugation. GFP nanobody-coupled Sepharose resin[62] was added to the lysate to pull down the TCR-CD3 complex via the GFP-fused CD3ζ subunits. The mixture was rotated at 4 °C for at least 1.5 hours. The resin was collected in a gravity column and washed with SEC buffer (50 mM Tris pH 8, 150 mM NaCl, 0.02% GDN). PreScission protease (Cytiva) and 0.5 mM DTT were added to the washed resin resuspended in SEC buffer and rotated overnight at 4 °C to cleave the TCR-CD3 complex off the GFP-bound resin. The flowthrough and additional subsequent washes of the resin were collected and concentrated in a 100 kDa MWCO Amicon Ultra centrifugal filter, then injected into a Superose 6 Increase 10/300 column (Cytiva) equilibrated to SEC buffer. Peak fractions were collected and concentrated in a 100 kDa MWCO Amicon Ultra centrifugal filter for cryoEM. TCR-CD3 complexes for SPR studies was purified in a similar fashion as above, with minor changes described below. Streptactin Superflow Plus resin (QIAGEN) was used for affinity purification. The strep-tagged complex was eluted from the resin using buffer containing 5 mM desthiobiotin (Sigma), followed by SEC.

### Preparation of MAGEA4 pMHC/2 M2 Fab and MAGEA8 pMHC/2M2 Fab complexes
For cryoEM studies of the PN45545 and PN45428 TCR-CD3 complexes with MAGEA4 pMHC, the pMHC protein was prepared by refolding of E. coli-expressed HLA-A2 and β2M inclusion bodies in the presence of MAGEA4 (230-239) peptide. Inclusion bodies (solubilized in Urea-containing buffer) and MAGEA4 peptide were diluted in refold buffer (100 mM Tris pH 8.0, 400 mM L-Arg pH 8.0, 2 mM EDTA, 5 mM reduced L-glutathione, 0.5 mM oxidized L-glutathione, 0.5 mM PMSF) and incubated at 4 °C with gentle agitation for four days. The reaction was concentrated using a Vivaflow 200 device (10 kDa MWCO, Sartorius) and Amicon Ultra centrifugal filters, then purified by SEC on a Superdex 75 gel filtration column. Peak fractions were collected and concentrated using 10 kDa MWCO Amicon Ultra centrifugal filters.

For cryoEM analysis of pMHCs in the absence of TCR and SPR studies, single-chain disulfide-stabilized forms[42] of MAGEA4 and MAGEA8 pMHCs were used. The constructs have the following design in which the peptide is stabilized by a disulfide bond between the linker cysteine and a cysteine introduced at position 84 of HLA-A2 (Y84C): Peptide-GCGGS-2x G$_4$S-β2M-4x G$_4$S-HLA-A2 Y84C (res. 1-276; amino acid numbering excludes N-terminal signal peptide)−2xMyc-

6xHis. The proteins were expressed in CHO-K1 cells and purified by immobilized metal affinity chromatography followed by SEC.

2M2 Fab was prepared from purified 2M2 mouse IgG1 antibody (RRID:AB_492835, BioLegend 316302) following standard protocols supplied in the Pierce Mouse IgG1 Fab Preparation Kit (Thermo Fisher). Complexes of refolded MAGEA4 pMHC bound to 2M2 Fab were isolated by mixing the two components and separating the complex by SEC using a Superdex 200 Increase column. This SEC-purified material was used to make complexes with TCR-CD3 for cryoEM.

## CryoEM sample preparation and data collection
CryoEM grids of PN45545 TCR-CD3 without antigen were prepared at a protein concentration of ~2 mg/mL. Complex of PN45545 TCR-CD3 and MAGEA4 pMHC/2M2 Fab was obtained by mixing the two components at concentrations of ~0.6 mg/mL and ~0.75 mg/mL, respectively, and incubating on ice prior to grid preparation. Complex of PN45428 TCR-CD3 and MAGEA4 pMHC/2M2 Fab was obtained by mixing the two components at concentrations of ~1.4 mg/mL and ~1.9 mg/mL, respectively. For TCR-free complexes of MAGEA4 pMHC/2M2 Fab and MAGEA8 pMHC/2M2 Fab, equal volumes of single-chain pMHC and 2M2 Fab at ~3 mg/mL were mixed and incubated on ice prior to grid preparation. 0.15% of PMAL-C8 amphipol (Anatrace) was added to the TCR-free pMHC/2M2 Fab samples immediately prior to grid preparation to aid vitrification. UltrAuFoil 1.2/1.3 grids were used for the unligated PN45545 TCR-CD3 sample and the TCR-free pMHC/2M2 Fab complex samples. UltrAuFoil 0.6/1 grids were used for the TCR-CD3 complexes with MAGEA4 pMHC. In each case, grids were freshly plasma cleaned in a Solarus II (Gatan) using a $H_2/O_2$ gas mixture. A Vitrobot Mark IV (Thermo Fisher) operated at 4 °C and 100% humidity was used for blotting the grids and plunge freezing them into liquid ethane cooled by liquid nitrogen.

Grids were loaded into a Titan Krios G3i electron microscope equipped with a BioQuantum K3 (Gatan). Images were collected in counted mode at a nominal magnification of 105,000x, yielding a pixel size of 0.85 Å. A defocus range of −1.4 to −2.4 μM was set for data collection using EPU (Thermo Fisher). The energy filter was inserted with slit width 20 eV. Each movie was dose-fractionated into 46 frames over a 2 second exposure and had a total dose of ~40 electrons per Å². Further details of the data collections leading to the structures are shown in Table 1.

## CryoEM data processing
CryoEM data were first processed using cryoSPARC v2[63] to assess data quality and generate initial 3D reconstructions. RELION 3[64,65] was used to determine the final maps using the same general workflow for each sample, summarized below. Details are shown in Supplementary Figs. 2, 3, 7, 8. Movies were dose-weighted, aligned, and summed using MotionCor2[66] as implemented in RELION. CTF parameters were estimated using gctf[67]. Micrographs with poor resolution estimates were removed from further processing. Laplacian-of-Gaussian picking was used on a subset of micrographs, followed by 2D classification to generate templates that were used for autopicking on the entire dataset. Multiple rounds of 2D classification and 3D classification were conducted to identify a homogenous subset of particles used for refinement. For the MAGEA4 pMHC/2M2 Fab and MAGEA8 pMHC/2M2 Fab datasets, focused 3D classification runs employing masks around the HLA molecule or the peptide groove region were conducted to identify particles with well-resolved density around the HLA-presented peptide. Particles were subjected to CTF refinement and Bayesian polishing to improve resolution. Map resolutions were calculated using RELION postprocessing. Maps that were filtered to their local resolution calculated in RELION were sharpened by phenix.auto_sharpen[68] for model building and visualization.

## Model building and refinement
Manual model building was conducted in Coot 0.8.9[69] and real space refinement of models was conducted using Phenix 1.19[70]. A previously published TCR-CD3 complex cryoEM structure (PDB ID: 6JXR)[28] was used as an initial model for building of the unligated PN45545 TCR-CD3 complex, which was in turn used as an initial model for the pMHC complexes of PN45545 TCR-CD3 and PN45428 TCR-CD3. A published crystal structure of HLA-A2/β2M/MAGEA4 (230-239) (PDB ID 1I4F)[40] was used as an initial model to build the pMHC structures. Atomic models for the 2M2 Fab fragment were not built because its sequence was not provided by the supplier. PyMOL[71], UCSF Chimera[72], and UCSF ChimeraX[73] were used to visualize models and maps. TCR docking angles were determined using the TCR3d database[16,74].

## T cell engineering
Data shown in Fig. 4a were obtained from primary human T cells expressing the MAGEA4 TCRs. Data shown in Supplementary Fig. 6 were obtained from engineered Jurkat cells expressing MAGEA4 TCRs.

Leukopaks were purchased from StemExpress (LE002.5F), drawn from a healthy female donor under the authority of their Institutional Review Board (IRB). Total T-cells were isolated by negative selection (Stem Cell Technologies #17951) and activated with CD3/CD28 Dynabeads (Life Technologies 11132D) in supplemented CTS OpTmizer (Life Technologies A1048501) media containing 4 mM glutamine, 10 mg/ml gentamicin (Life Technologies 15710064), 100 U/ml hIL-2 (Miltenyi 130-097-748), and 10 ng/ml hIL-15 (Miltenyi 130-095-765).

TCRs were expressed in primary human T cells by targeting AAV-encoded TCR constructs to the *TRAC* locus as previously described[30]. Three days after activation, beads were removed, and cells were nucleofected with Crispr RNP consisting of Cas9 protein (Life Tech A36499) complexed with a mixture of modified synthetic guide RNAs (sgRNAs, IDT) targeting the TRAC (GCUGGUACACGGCAGGGUCA) and TRBC1/2 (UGGGAAGGAGGUGCACAGUG) genes in their first exons. 5e6 T-cells were suspended in 100 μl nucleofection buffer (Lonza VPA-1002) containing 30 μg Cas9 complexed with 150 pmol of each sgRNA, and electroporated with the T-020 program on the Lonza Nucleofector IIb. Cells were transferred immediately into media containing adeno-associated virus (AAV, 4e4 viral genomes/cell) vectors encoding homology directed repair templates for TRAC insertion. Every 2–3 days, cells were diluted to 0.5–1e6 cell/ml in media with fresh cytokines. TCR expression and antigen binding was evaluated by flow analysis with pHLA tetramers.

To express customized MHC-I-restricted TCRs in Jurkat cells (original Jurkat E6 cells from ATCC TIB-152), endogenous *TRA* and *TRB* genes were disrupted via Crispr/Cas9, and *CD8A* and *CD8B* (single cistron, formatted as CD8A-F2A-CD8B) were stably introduced via lentiviral transduction (pLVX.EF1a.IRES.puro; Takara 631988). Modified Jurkat cells were then transduced with lentiviral vectors (pLVX.EF1a.IRES.Zeocin) encoding TCRs (single cistron, formatted as TCRA-F2A-TCRB). Jurkat cells were cultured in supplemented RPMI 1640 (Irvine Scientific 9160) media containing 10% FBS (Avantor Seradigm 97068-085), 1X Penicillin-Streptomycin-Glutamine (Life Technologies 10378016), and selection with 1 μg/mL puromycin (Sigma-Aldrich P8833) and 200 μg/mL zeocin (Life Technologies R25001).

## Flow cytometry
A total of 10^6 engineered primary human T cells expressing MAGE-A4 TCRs were first stained with either MAGE-A4 (GVYDGREHTV) HLA-A*02:01 Custom Tetramer (PE, MBL #TBCM3-A021I-1) or MAGE-A8 (GLYDGREHSV) HLA-A*02:01 Custom Tetramer (PE, MBL #TBCM3-A021I-1) diluted 1:10 in 50 uL fluorescence-activated cell sorting (FACS) buffer (1X phosphate-buffered saline (PBS), 5% fetal bovine serum) and incubated at 4 °C for 20 minutes. Staining antibodies against human CD3ε (RRID:AB_2744387, Brilliant Ultraviolet 395, cl. UCHT1, BD 563546, final dilution 1:50), human CD8α (RRID:AB_2561942, Brilliant

Violet 510, cl. RPA-T8, BioLegend 301048, final dilution 1:200), and human CD4 (RRID:AB_314080, PE-Cyanine7, cl. RPA-T4, BioLegend 300512, final dilution 1:200) were suspended in 50 uL Brilliant Stain Buffer (BD 566349) and added to wells for a final staining volume of 100 uL. Cells were incubated for an additional 20 minutes at 4 °C and then washed twice with FACS buffer. SYTOX AADvanced Dead Cell Stain (Invitrogen S10349) was diluted 1:2500 in PBS containing 5 mM EDTA (Invitrogen 15575020), then added to cells at a final dilution of 1:5000 prior to analysis on the BC CytoFLEX LX flow cytometer. Flow cytometry data were acquired using CytExpert and analyzed using FlowJO 10.5.2. At least 30,000 and 50,000 live, single cells were acquired for each sample shown in Fig. 4a and Supplementary Fig. 6, respectively.

To generate customized pMHC tetramers for flow cytometry, MAGEA4 (GVYDGREHTV), MAGEA8 (GLYDGREHSV), MAGEA4 V2L (GLYDGREHTV), and MAGEA4 T9S (GVYDGREHSV) peptides were purchased from Celtek Bioscience (Franklin, TN). Lyophilized peptides were resuspended in dimethyl sulfoxide (Sigma-Aldrich D2650) at a concentration of 7.5 mg/mL. Each custom peptide was loaded onto QuickSwitch™ HLA-A*02:01 Tetramer (PE, MBL TB-7300-K1), made from monomer units folded with an irrelevant exchangeable peptide, per the manufacturer's instructions. Briefly, the custom peptide was further diluted in water to 1 mM, and 1 μL of the diluted peptide was mixed with 50 μL QuickSwitch™ HLA-A*02:01 Tetramer and 1 μL Peptide Exchange Factor for a final peptide concentration of 20 μM. The peptide exchange reaction was allowed to proceed for at least 4 hours at room temperature protected from light, after which the tetramer was ready for use in flow analysis to evaluate antigen binding to TCRs. Gating strategies for flow cytometry experiments are shown in Supplementary Fig. 9 and 10.

### Surface Plasmon Resonance (SPR)

SPR affinity analysis was performed on a Cytiva T-200 instrument using T200 Control Software version 3.2.1 (Cytiva) for data acquisition. A CM5 sensor chip (Cytiva) was prepared by EDC/NHS coupling of Strep-Tactin-XT (IBA Lifesciences). Running buffer was 8 mM TRIS, 7 mM HEPES, 150 mM NaCL, 0.067% GDN, pH 7.2. Approximately 3000RU of PN45428 TCR-CD3 or 1600RU of PN45545 TCR-CD3 were immobilized. MAGEA4 (230-239) or MAGEA8 (232-241) pMHC samples (single-chain disulfide-stabilized) were prepared by 2-fold, 8 point serial dilution of 100 μM stock solutions. MAGEA4 and MAGEA8 pMHC samples were injected at 50uL/min for 60 s. Double-referenced binding responses were measured prior to the end of injections. Steady-state affinity analysis was performed using Scrubber v 2.0c (BioLogic Software) Rmax were floated and fit to the data.

### Reporting summary

Further information on research design is available in the Nature Portfolio Reporting Summary linked to this article.

## Data availability

Regeneron materials described in this manuscript may be made available to qualified, academic, noncommercial researchers through a materials transfer agreement upon request at https://regeneron.envisionpharma.com/vt_regeneron/. For questions about how Regeneron shares materials, use the email address preclinical.collaborations@regeneron.com. Structural coordinates have been deposited to the Protein Data Bank (PDB) and cryoEM maps have been deposited to the Electron Microscopy Data Bank (EMDB) with accession numbers 8ES7 and EMD-28570 (PN45545 TCR-CD3), 8ES8 and EMD-28571 (PN45545 TCR-CD3 in complex with MAGEA4 pMHC), 8ES9 and EMD-28572 (PN45428 TCR-CD3 in complex with MAGEA4 pMHC), 8ESA and EMD-28573 (MAGEA4 pMHC in complex with 2M2 Fab), 8ESB and EMD-28574 (MAGEA8 pMHC in complex with 2M2 Fab). Source data for Figs. 4b and 4c are provided with this paper. Protein Data Bank (PDB) codes for additional structures used in this analysis are 6JXR, 1I4F and 7PHR. Source data are provided with this paper.

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

## Acknowledgements
We thank Luke McGoldrick and David DiLillo for critical reading of the manuscript, Nam Nguyen for assistance with cell culture and protein expression, Annabel Romero Hernandez for discussions, Hueiwen Tan and Douglas MacDonald for project management, and the Regeneron cloud/HPC teams for supporting cryoEM data storage and processing.

## Author contributions
K.S., D.D., M.J.M., A.R., G.D.Y., A.J.M., J.C.L., W.C.O., and M.C.F. conceptualized the studies. K.S. and J.J. expressed and purified proteins and prepared cryoEM samples. K.S. acquired and processed cryo-EM data and built the atomic models, with contributions from Y.Z. and M.C.F. K.C. and M.J.M. conducted flow cytometry studies. D.D. and A.R. conducted SPR studies. G.D.Y., A.J.M., J.C.L., W.C.O., and M.C.F. analyzed data and supervised the overall project. K.S., M.J.M., and D.D. drafted the manuscript with contributions from Y.Z., W.C.O., and M.C.F. The manuscript was finalized by all authors.

## Competing interests
All authors own options and/or stock of Regeneron. G.D.Y., A.J.M., J.C.L., and W.C.O. are officers of Regeneron.
