## [Peer Review File · Nature Communications]

Structural analysis of cancer-relevant TCR-CD3 and peptide-MHC complexes by cryoEMREVIEWER COMMENTS

Reviewer #1 (Remarks to the Author):

Saotome and colleagues report the cryo-EM structures of a pair of therapeutically relevant MAGEA4-targeting T-cell receptors (TCRs) bound to their cognate peptide-human leukocyte antigen (peptide-HLA) molecules, revealing important structural principles underpinning antigen recognition by different TCRs. They perform these analyses on full-length, hetero-octameric CD3-TCR complexes, furthering our understanding of the structure of this large membrane protein complex. Moreover, they determine the structures of both MAGEA4 and MAGEA8 tumor antigens bound to HLA, yielding insights to the structural chemistry of antigen discrimination in this system. Overall, I believe this work to be sufficiently novel, broadly important, and technically strong, and thus recommend publication with the following revisions:

1) The authors spend much time and effort to emphasize the importance of their work in establishing the use of cryo-EM for the structural analysis of MHC antigens and their complexes with TCR. The applicability of cryo-EM to this question is not nearly as surprising as they seem to think, and indeed Susac and colleagues have recently already published the cryo-EM structure of a complete CD3-TCR bound to peptide-HLA (PMID: 35985289 DOI: 10.1016/j.cell.2022.07.010). The authors should therefore greatly deemphasize this aspect of their work.

2) The authors should discuss their findings in the context of the recently published structure. Grossly, it appears that the structures show that the complexes have a remarkably similar architecture, and their conclusions would be consistent. Moreover, it provides further support for the authors' conclusion that their work "supports the notion that the overall structure and assembly of TCR-CD3 is unaffected by differences in TCR variable region sequence."

3) The authors' work highlights the remarkably different binding angles tolerable for a TCR-peptide-HLA ligation ("characterized by docking angles of 45° and 94° for PN45545 and PN45428, respectively"). The authors should clarify whether they both bind with the same polarity (it appears that they do) and whether both would be consistent with the co-receptor binding constraints proposed by Susac et al. As the authors' protein is fully glycosylated, their pair of structures suggests flexibility in TCR-HLA interaction geometries, which is likely of physiologic import.

4) The impact of two relatively conservative point mutations on the HLA binding geometry and TCR reactivity of MAGEA4 vs. MAGEA8 is striking, especially given the fact that neither MAGEA4 residue contacts the TCR in their structures. They provide a structural argument that the altered peptide geometry and reactivity for MAGEA8 is the effect of V2L on D4 positioning, with no proposed role for T9S. Although their model is plausible, the density into which this peptide model is built is not definitive (Fig 5e,f and extended data Fig 5d and 6d). While it may not be possible to overcome this limitation in the raw data, their argument for the import of V2L would be strengthened by performing their SPR or flow cytometry experiments with the "hemi-mutant" MAGEA4/8 hybrids V2L and T9S separately. As the authors point out, the use of the 2 positions in peptide engineering to increase HLA affinity makes this an important conclusion of their paper.

5) The Fig 1 c/d legend refers to dotted lines, but no dotted lines seem to be present.

6) The number of replicates for the SPR experiments should be listed, as means and SEM are given.

Reviewer #2 (Remarks to the Author):

This is an important manuscript describing the application of cryoEM to determining

high-resolution structures of full-length TCR/CD3/pMHC complexes, as well as of pMHCs alone. The authors demonstrate that cryoEM can yield atomic-level information on TCR/pMHC interactions that up to now could only be gotten from X-ray crystallography. This study builds on a previous cryoEM structure determination of an unbound TCR/CD3 complex by Dong et al. It also coincides with the recently reported structure of a different TCR/CD3/pMHC complex from the Simon Davis lab (Cell 185, 3201 (2022)). The current manuscript was presumably submitted, or at least written, before publication of the Davis paper. Either way, the two studies are complementary, and one does not detract from other. However, Saotome et al. must include the Davis structure in any revision.

Points to address:

1. The authors need to provide some additional EM information. For instance, they did not show maps for the TCR/CD3/pMHC complexes while zooming onto the peptide. In Fig. 2b and 2e, it looks like a carved volume around the peptide. The authors should show this to enable an evaluation of peptide quality in the map. In Extended Data Figs. 3d and 4d, they should show zoomed out panels highlighting exactly which part of the complexes is being depicted.
2. How does the location of the putative cholesterol molecule in the PN45545 TCR/CD3 structure compare with that of cholesterol in the Davis TCR/CD3/pMHC structure?
3. In Fig. 1e, the authors should include the Davis TCR/CD3 complex in the superpositions.
4. In Extended Data Fig. 1, there are several bands of ~35 kDa that do not appear to correspond to TCR or CD3 subunits. What could these bands be?
5. In the Davis study, an engineered high-affinity (nanomolar) TCR was used, presumably to stabilize the TCR/CD3/pMHC complex for cryoEM. However, Saotome et al. were able to use a wild-type TCR having only micromolar affinity, which is very relevant to extending their approach to other TCR/CD3/pMHC complexes. The authors should emphasize this advantage.
6. For cryoEM analysis of unbound pMHCs, Saotome et al. used single-chain, disulfide-stabilized versions of MAGEA4/HLA-A2 and MAGEA8/HLA-A2. Why did they do this, given that unaltered MAGEA4/HLA-A2 was used for TCR/CD3/pMHC structure determinations?
7. In the Discussion, the authors note that they did not observe and structural changes in the TCR/CD3 complex upon binding pMHC, in agreement with the Davis study. In addition to mechanotransduction as a possible mechanism to explain TCR triggering, they should also mention dynamic allostery (see J. Biol. Chem. 295, 914 (2020) for a recent review).

The reviewers' comments on our manuscript are reproduced below verbatim. We have inserted our responses to the reviewers at the appropriate places, with our text highlighted in yellow. Please let us know if this highlighting has not been preserved properly.

REVIEWER COMMENTS

Reviewer #1 (Remarks to the Author):

Saotome and colleagues report the cryo-EM structures of a pair of therapeutically relevant MAGEA4-targeting T-cell receptors (TCRs) bound to their cognate peptide-human leukocyte antigen (peptide-HLA) molecules, revealing important structural principles underpinning antigen recognition by different TCRs. They perform these analyses on full-length, hetero-octameric CD3-TCR complexes, furthering our understanding of the structure of this large membrane protein complex. Moreover, they determine the structures of both MAGEA4 and MAGEA8 tumor antigens bound to HLA, yielding insights to the structural chemistry of antigen discrimination in this system. Overall, I believe this work to be sufficiently novel, broadly important, and technically strong, and thus recommend publication with the following revisions:

1) The authors spend much time and effort to emphasize the importance of their work in establishing the use of cryo-EM for the structural analysis of MHC antigens and their complexes with TCR. The applicability of cryo-EM to this question is not nearly as surprising as they seem to think, and indeed Susac and colleagues have recently already published the cryo-EM structure of a complete CD3-TCR bound to peptide-HLA (PMID: 35985289 DOI: 10.1016/j.cell.2022.07.010). The authors should therefore greatly deemphasize this aspect of their work.

Thank you for this comment. Our manuscript was prepared and finalized for submission before publication of the Susac et al manuscript, which is why it was not cited. We have added citations to Susac et al in the results and discussion sections (following Reviewer #1 comments 2 and 3) below.

Even considering Susac et al, we believe our manuscript makes important inroads towards establishing cryoEM for TCR/pMHC complex analysis. First, we used typical micromolar-affinity TCRs for our study, while Susac et al used an affinity-enhanced picomolar TCR. We believe this to be important because most TCR/pMHC interactions are inherently low affinity (note that Reviewer #2 suggested that we emphasize this point). Second, in addition to TCR/pMHC complexes, we have determined cryoEM structures of pMHC complexes alone. We think our finding that a commercially available anti B₂M Fab makes pMHC (and presumably, other antigen-presenting molecules containing B₂M) sufficiently large for cryoEM will be of value to the field. We therefore are reluctant to greatly de-emphasize the technical aspects of our work. Nonetheless, we have made some changes to the introduction and discussion sections to tone down some claims of novelty in light of the Susac et al publication.

2) The authors should discuss their findings in the context of the recently published structure. Grossly, it appears that the structures show that the complexes have a remarkably similar architecture, and their conclusions would be consistent. Moreover, it provides further support for the authors' conclusion that their work "supports the notion that the overall structure and assembly of TCR-CD3 is unaffected by differences in TCR variable region sequence."

Thank you for this suggestion. We have added the Susac et al structure (PDB 7PHR) to the structural alignment of Figure 2d and cited the paper in the associated portion of the results section.

3) The authors' work highlights the remarkably different binding angles tolerable for a TCR-peptide-HLA ligation ("characterized by docking angles of 45° and 94° for PN45545 and PN45428, respectively"). The authors should clarify whether they both bind with the same polarity (it appears that they do) and whether both would be consistent with the co-receptor binding constraints proposed by Susac et al. As the authors' protein is fully glycosylated, their pair of structures suggests flexibility in TCR-HLA interaction geometries, which is likely of physiologic import.

Thank you for this suggestion. We have clarified in the text that both TCRs bind with the same, canonical polarity using different docking angles. We have also annotated Fig 2c,f with the TCR docking angles calculated from TCR3d database. We have also added Supplementary Fig. 5, which models how CD8 binds our MAGEA4 TCR-CD3 complexes, and added the following text: "To assess CD8 co-receptor binding geometry, we used a published crystal structure of murine CD8αβ bound to MHC³⁹ to model binding of CD8αβ in the context of both full-length MAGEA4 TCR-CD3 complexes (Supplementary Fig. 5). Despite their distinct docking angles, both PN45545 and PN45428 bind MAGEA4 pMHC such that the CD8αβ Ig domain C-termini would be oriented toward the T cell membrane, favoring a *cis* configuration of TCR/CD3/CD8 and *trans* binding of pMHC/CD8 as proposed previously^{29,39}. This observation illustrates how the geometric constraints of coreceptor binding could accommodate a range of TCR/pMHC binding angles."

4) The impact of two relatively conservative point mutations on the HLA binding geometry and TCR reactivity of MAGEA4 vs. MAGEA8 is striking, especially given the fact that neither MAGEA4 residue contacts the TCR in their structures. They provide a structural argument that the altered peptide geometry and reactivity for MAGEA8 is the effect of V2L on D4 positioning, with no proposed role for T9S. Although their model is plausible, the density into which this peptide model is built is not definitive (Fig 5e,f and extended data Fig 5d and 6d). While it may not be possible to overcome this limitation in the raw data, their argument for the import of V2L would be strengthened by performing their SPR or flow cytometry experiments with the "hemi-mutant" MAGEA4/8 hybrids V2L and T9S separately. As the authors point out, the use of the 2 positions in peptide engineering to increase HLA affinity makes this an important conclusion of their paper.

We have conducted flow cytometry experiments on the suggested V2L and T9S "hemi-mutant" peptides (now in Supplementary Fig. 6). Consistent with our proposed mechanism, the V2L mutant showed reduced staining relative to MAGEA4 while the T9S mutant did not. Therefore, the peptide selectivity for MAGEA4 over MAGEA8 of these TCRs is indeed a result of the V2L substitution. Thank you for the suggestion for this experiment because we believe it strengthens our manuscript.

5) The Fig 1 c/d legend refers to dotted lines, but no dotted lines seem to be present.

Thank you for catching this. An earlier version of the figure had dotted lines, which were removed by accident while formatting for submission. The dotted lines have been added back to Fig 1c/d.

6) The number of replicates for the SPR experiments should be listed, as means and SEM are given.

The experiments were run in duplicate. This is now added to the figure 4 caption.

Reviewer #2 (Remarks to the Author):

This is an important manuscript describing the application of cryoEM to determining high-resolution structures of full-length TCR/CD3/pMHC complexes, as well as of pMHCs alone. The authors demonstrate that cryoEM can yield atomic-level information on TCR/pMHC interactions that up to now could only be gotten from X-ray crystallography. This study builds on a previous cryoEM structure determination of an unbound TCR/CD3 complex by Dong et al. It also coincides with the recently reported structure of a different TCR/CD3/pMHC complex from the Simon Davis lab (Cell 185, 3201 (2022)). The current manuscript was presumably submitted, or at least written, before publication of the Davis paper. Either way, the two studies are complementary, and one does not detract from other. However, Saotome et al. must include the Davis structure in any revision.

Points to address:

1. The authors need to provide some additional EM information. For instance, they did not show maps for the TCR/CD3/pMHC complexes while zooming onto the peptide. In Fig. 2b and 2e, it looks like a carved volume around the peptide. The authors should show this to enable an evaluation of peptide quality in the map. In Extended Data Figs. 3d and 4d, they should show zoomed out panels highlighting exactly which part of the complexes is being depicted.

Thank you for this suggestion. Indeed, Fig 2b and 2e are carved volumes around the peptide. We have added panels to Supplementary Figs 3 and 4 showing zoomed-in views of the peptide region in the map. We have also added figure panels showing zoomed-out views.

2. How does the location of the putative cholesterol molecule in the PN45545 TCR/CD3 structure compare with that of cholesterol in the Davis TCR/CD3/pMHC structure?

The putative cholesterol-like molecule in our structure appears to be in the same location as the density in the Davis structure (PDB 7PHR), as well as structures in Chen et al *Mol Cell* 2022. We now make note of this in the text.

3. In Fig. 1e, the authors should include the Davis TCR/CD3 complex in the superpositions.

Thank you for this suggestion, we edited Fig 1e to include their structure.

4. In Extended Data Fig. 1, there are several bands of ~35 kDa that do not appear to correspond to TCR or CD3 subunits. What could these bands be?

We can't say conclusively what the bands are since we did not assess these bands by mass spectrometry. We speculate they are impurities carried through the prep, or partially reduced/denatured CD3 dimers.

5. In the Davis study, an engineered high-affinity (nanomolar) TCR was used, presumably to stabilize the TCR/CD3/pMHC complex for cryoEM. However, Saotome et al. were able to use a wild-type TCR having only micromolar affinity, which is very relevant to extending their approach to other TCR/CD3/pMHC complexes. The authors should emphasize this advantage.

Thank you for noting this, we also believe that this is an advantage and have now alluded to this in the introduction as well as the final paragraph of the discussion section.

6. For cryoEM analysis of unbound pMHCs, Saotome et al. used single-chain, disulfide-stabilized versions of MAGEA4/HLA-A2 and MAGEA8/HLA-A2. Why did they do this, given that unaltered MAGEA4/HLA-A2 was used for TCR/CD3/pMHC structure determinations?

The reason we did this is reagent availability. We had single chain reagents available for both MAGEA4 and MAGEA8 pMHCs and refolded material for only the MAGEA4 pMHC (which was used for the TCR-CD3 complexes). To minimize confounding variables, we used the single chain reagents for the cryoEM analysis of both MAGEA4 and MAGEA8 pMHCs. We also have clarified in the methods section that the single chain reagents were used in our SPR experiments, validating their functionality (Fig 4b,c).

7. In the Discussion, the authors note that they did not observe and structural changes in the TCR/CD3 complex upon binding pMHC, in agreement with the Davis study. In addition to mechanotransduction as a possible mechanism to explain TCR triggering, they should also mention dynamic allostery (see J. Biol. Chem. 295, 914 (2020) for a recent review).

Thank you for pointing us to this nice review. We now mention the possibility of dynamic allostery in the discussion section and cite the review.